# Exploratory Analysis of Related Factors with Absenteeism and Presenteeism on Workers: Using the Fourth Korea Working Condition Survey

**DOI:** 10.3390/ijerph182111214

**Published:** 2021-10-26

**Authors:** Young-Hyeon Bae

**Affiliations:** Korea National Rehabilitation Center, Department of Healthcare and Public Health, National Rehabilitation Research Institute, Seoul 01022, Korea; researcher2018@korea.kr

**Keywords:** Korea working condition survey, absenteeism, presenteeism, relationship

## Abstract

Purpose: This study aimed to identify the factors related to absenteeism and presenteeism in workers and to provide basic evidence to help improve their quality of life and work productivity. Methods: Data from the Fourth Korea working condition survey were analyzed. A stepwise regression model was developed to identify the related factors for exploratory analysis. Results: Absenteeism and subjective risk perception related to work and health conditions were the strongest predictive factors, followed by presenteeism. Fatigue was the strongest predictive factor for presenteeism, followed by a high temperature at the workplace, lower back pain, and other health conditions, in that order. Conclusion: The quality of life and the productivity of workers could be improved by focusing on the factors affecting absenteeism and presenteeism, such as the working environment and health status.

## 1. Introduction

Absence from work refers to a failure to show up at work or the place designated by the business owner on a workday, and is affected independently or interactively by factors such as disease, accident, injury, and personal matters [1]. Absence is considered a relatively easy indirect indicator of a worker’s health state and work productivity. Absence due to a work-related health problem is referred to as absenteeism [2].

Despite having work-related health problems, some workers may force themselves to work because their health problems do not require immediate treatment or surgery and they may be disadvantaged at work by sickness absence. The loss of productivity caused by workers presenting to work despite having a health problem is known as presenteeism [3].

Presenteeism occurs when workers are present at work, which is the opposite of absenteeism. These two have been argued to have a reverse proportional relationship, in which one decreases when one increases. Although absenteeism can be easily and objectively measured, this is not the case with presenteeism. Because it is difficult to estimate the impact and loss caused by workers presenting to work despite a health problem simply based on the absence rate, it has been suggested that both factors must be considered when discussing work productivity [1].

Absenteeism, which incurs indirect costs such as disability from an accident, use of a healthcare institution, and direct loss of working hours, accounts for 40% of productivity costs, but presenteeism, which incurs indirect costs from reduced work ability, accounts for 60% of these costs [1]. In addition, the cost resulting from presenteeism in the United States alone exceeds USD 180 billion a year, which surpasses that from absenteeism (USD 118 billion). Thus, a strong consensus has been reached regarding the fact that presenteeism leads to a greater loss of productivity than absenteeism [4]. This is also interpreted as an iceberg effect, where the visible aspect of work loss (absenteeism) is reduced by the part under the water (presenteeism) [4].

Previous studies have indicated several determinants of presenteeism including organizational factors as understaffing, low organizational support, and strict absence policy and work-related factors such as workload and time pressure, as well as individual factors such as affective commitment and financial difficulties [5]. A high prevalence of presenteeism is concerning, given that it could lead to health problems and absenteeism in later periods [6]. Previous studies have reported that the experience of presenteeism is related to poor self-rated health, depression, absenteeism, and lower work performance [7,8,9,10].

The Korea Occupational Safety and Health Agency (KOSHA) has benchmarked the European Working Conditions Survey and has adopted a Korean version of the Working Conditions Survey (KWCS). The first and second surveys were conducted in four-year cycles, and the cycle was changed to one year in the third survey [11]. However, considering the nature of the survey and the sensitivity of the analytical results, the cycle was changed to three years from the fourth survey [11]. The KWCS is conducted through interviews with employees and self-employed individuals regarding key issues about work, employment, and occupational health and safety, to present results about working conditions. In the second to fourth KWCS, no trend of absenteeism was observed, but presenteeism tended to increase over the years. In the 2014 KWCS, medical absenteeism was 8.8%, while presenteeism was 25.8% [11].

Recently, business owners or health managers in Korea have grown more interested in the indirect losses caused by presenteeism, as opposed to the direct and visible causes, such as absence rates and direct medical costs when measuring work productivity. However, although there have been studies on absenteeism and presenteeism in specific groups of people, almost no studies have been conducted that comprehensively analyze the factors that influence the overall working population [12].

Therefore, this study aimed to explore the factors that affect absenteeism and presenteeism in the overall working population using data from the KWCS conducted by KOSHA to present foundational data for improving workers’ quality of life and work productivity. The study was approved by our Institutional Review Board.

## 2. Methods

The 2014 KWCS was conducted from June to September 2014 through a face-to-face interview with 50,007 workers in 17 cities and provinces nationwide. The KWCS is publicly available at http://www.kosha.or.kr/ (accessed on 25 June 2018) under permission from Korea Occupational Safety and Health Agency.

The numbers in the KWCS are not the investigation results from the population but the estimated numbers based on a sample of representative workers [12]. The survey population was a nationally representative sample of the economically active population (≥15 years old) including waged workers, self-employed/employer, unpaid family workers, and others.

The 2014 KWCS was the fourth survey conducted with 226,092 people after the exclusion of 176,085 workers who refused to participate and those who met the exclusion criteria. The size of this study population was 50,007 and included workers who participated in the on-site survey about working and employment conditions (Table 1).

The contents of the survey were developed by adding various items related to occupational health and safety to the items used in the European Working Conditions Survey. The collected data reflect workers’ perspectives, features of their organizations, and their homes. To create a map of changes—generally regarding working life—that occur over time, diverse topics were added and reviewed. The survey consists of current physical factors, socio-psychological risks, leadership, changes at work, work-life balance, flexibility and flexicurity, and modern forms of working organizations.

To ensure a comprehensive exploration of the factors that affect absenteeism and presenteeism, this study selected and analyzed variables related to the working environment and health in the fourth KWCS data (Appendix A).

To explore the factors that affect absenteeism and presenteeism, Q72 “How many days have you been absent from work in the past 12 months because of a health problem?” was set as a dependent variable and the remaining variables were set as independent variables for the absenteeism model. For the presenteeism model, Q74 “Have you ever come to work even when you were sick in the past 12 months?” was set as a dependent variable and the remaining variables were set as independent variables.

Statistical analyses were performed using SPSS version 24.0. Spearman correlation coefficients were used to identify the variables that correlated with absenteeism and presenteeism. Next, a stepwise regression analysis was used to identify the independent variables that significantly affected the dependent variables—absenteeism and presenteeism—by entering the variables that were found to be significantly correlated with them in the previous analysis.

The correlation coefficient was confirmed by 0.9 or more and the variance expansion factor value was confirmed by 2 or more to verify the multicollinearity.

## 3. Results

### 3.1. Variables Significantly Correlated with Absenteeism and Presenteeism

The correlation analysis showed that the following question items were significantly correlated with absenteeism (range of correlation coefficients = 0.033–0.162): KQ4, Q18, Q23-A, Q23-C, Q23-D, Q23-E, Q23-G, Q24-A, Q24-C, Q24-F, Q24-G, Q24-H, Q24-I, Q30, Q33, Q36, Q45-A, Q45-B, Q66, Q67, Q68, Q69-A, Q69-C, Q69-D, Q69-E, Q69-H, Q69-I, Q69-J, Q69-K, Q69-M, Q69-N, and Q74 (Table 2).

The correlation analysis showed that the following question items were significantly correlated with presenteeism (range of correlation coefficients = 0.011–0.226): KQ4, Q18, Q20, Q23-A, Q23-B, Q23-C, Q23-D, Q23-E, Q23-F, Q23-G, Q23-H, Q24-A, Q24-B, Q24-C, Q24-D, Q24-E, Q24-F, Q24-G, Q24-H, Q24-I, Q30, Q33, Q36, Q44-A, Q44-B, Q45-A, Q45-B, Q66, Q67, Q68, Q69-A, Q69-B, Q69-C, Q69-D, Q69-E, Q69-F, Q69-G, Q69-H, Q69-I, Q69-J, Q69-K, Q69-L, Q69-M, Q69-N, and Q72 (Table 2).

### 3.2. Factors That Affect Presenteeism

The most significant predictor of presenteeism was “Have you had any of the following health problems in the past 12 months?—L. General fatigue” (R^2^ = 7.7%), followed by “How much are you exposed to the following while working?—Inhalation of smoke, fumes (welding or exhaust), powder, dust (e.g., wood dust, mineral dust)” (R^2^ = 2.1%), “How much are you exposed to the following while working?—low temperature, whether indoors or outdoors” (R^2^ = 1.7%), “Have you had any of the following health problems in the past 12 months?—C. Low back pain” (R^2^ = 0.9%), and “Have you had any of the following health problems in the past 12 months?—N. Other” (R^2^ = 0.9%). These variables explained 13.4% of the R^2^ variance (Table 3).

### 3.3. Factors That Affect Absenteeism

The most significant predictor of absenteeism was “Have you had any of the following health problems in the past 12 months?—J. Injury (by accident)” (R^2^ = 3.1%), followed by “How is your general health state?” (R^2^ = 0.7%), “Are any of the following included in your work?—I. Use the internet/e-mail for work” (R^2^ = 0.3%), “How often is the following type of situation included in your work?—B. Work under a strict deadline” (R^2^ = 0.2%), “How much are you exposed to the following while working?—Inhalation of smoke, fumes (welding or exhaust), powder, dust (e.g., wood dust, mineral dust)” (R^2^ = 0.3%), and “Have you ever come to work even when you were sick in the past 12 months?” These variables explained 4.8% of the R^2^ variance (Table 4).

## 4. Discussions

In 2002, Lockheed Martin assessed the impact of 28 health-related conditions on presenteeism. The findings revealed that even workers with non-serious conditions tended to show presenteeism symptoms and that companies recorded an approximately USD 3.4 million loss because of these 28 health conditions affecting presenteeism [13]. In a study on absenteeism and presenteeism related to migraines in more than 80 thousand workers of a large financial service company, Burton et al. (1999) found that the estimated loss due to absenteeism was USD 21.5 million and that from presenteeism was USD 24.4 million [14].

Increased presenteeism has a serious toll on a firm’s profits because of rising healthcare expenditure and reduced productivity. Moreover, the medical costs incurred by presenteeism account for 63% of the total indirect medical costs, which are 10 times higher than that caused by absence and three times higher than direct medical costs [13]. Burton et al. (1999). reported that although costs incurred by loss of working hours due to the use of healthcare facilities, disability from an accident, and direct loss of working hours were only about 40% of the total, indirect costs incurred by presenteeism, such as reduced work capability reached 60%, confirming that loss from presenteeism is greater than that caused by a loss of working hours [14]. Thus, Hemp (2004) presented the potential costs of presenteeism and argued that it is important to implement measures to lower presenteeism to increase productivity and reduce the costs of productivity loss [13]. In the 2014 KWCS, medical absenteeism was 8.8%, while presenteeism was 25.8% [5].

Our study shows that health problems due to general fatigue had the greatest impact on presenteeism, followed by the inhalation of smoke, fumes, powder, or dust, exposure to low temperatures indoors/outdoors, health problems due to low back pain, and other health problems in 2014 KWCS. Goetzel et al. (2004) compared the costs of absenteeism and presenteeism pertaining to ten physical and mental health states that incur the greatest costs [9]. They found that cardiovascular diseases such as hypertension, coronary artery disease, and heart disease, musculoskeletal disorders, such as low back pain and arthritis, bronchial diseases, such as cold, flu, and asthma, and mental diseases, such as depression and stress, are the main causes of presenteeism. Chronic disease, such as headache, allergy, arthritis, and asthma, and most mental health problems, such as depression, were found to incur losses from presenteeism [15]. Jeon et al. (2009) reported that the number of health problems a worker suffered correlated positively with presenteeism, meaning that presenteeism increased with the number of health problems [16]. Although the variables measured in previous studies and our study differed, health problems still had the most significant effects.

In a study on nurses, the most significant predictor of presenteeism was the number of health problems, and other significant predictors were age, the number of nurses working in the same ward, working in three shifts, and dissatisfaction with wages [12]. Among health problems, shoulder, low back, and neck pain, as well as fatigue and swelling of the feet had the most significant effects on presenteeism, and work environments in which nurses must work on their feet and continually move around or use their body to move or support patients were identified as relevant factors [1]. In a presenteeism study on physical therapists, the most significant variable was quality of life, followed by pain rating scale, nursing hospital (classification of care facilities), age, pediatric care (classification of department), daytime work, internship (type of work), and gender [17,18]. In addition, a poor work environment was reported to induce uncontrollable stress throughout the entire working period and to have adverse effects on productivity [19]. Similarly, in the present study, exposure to a negative work environment in addition to health problems had an impact on presenteeism. However, our participants and variables differed from those used in previous studies, so there were differences in the specific types of health problems and negative work environments that affected presenteeism.

This study explored the factors that affect absenteeism among workers and found that health problems caused by accidental injuries had the greatest effect, followed by perceived overall health, use of internet and emails for work, working under a strict deadline, inhalation of smoke, fumes, powder, or dust, and presenteeism. A previous study found that factors that affected absenteeism were mostly related to the working environment, including work position, shift work, number of days working at night, autonomy, degree of effect of one’s work on health, and social support [1]. Furthermore, although it is difficult for the worker to perceive the loss of work with only one or two health problems, loss of work increased with the number of health problems, inducing presenteeism and ultimately affected absence as well [19,20]. Although health problems and negative work conditions have commonly found to be predictors in this and previous studies, it is difficult to directly compare our findings with previous findings because of differences of subjects and methods. In addition, considering that presenteeism was found to affect absenteeism in our study as well as with other studies, it is difficult to estimate the influence and loss incurred by workers presenting to work even with a health problem, based solely on the absence rate. These findings support an argument made in a previous study that both presenteeism and absenteeism should be considered to analyze work productivity loss [12,21].

However, in previous studies, the loss of productivity from mental diseases, such as depression, outweighed that caused by chronic physical diseases. It was reported that depression and anxiety symptoms are more consistently associated with presenteeism than absenteeism, but in our study, depression and anxiety did not affect presenteeism. This prevented our findings from being directly compared to previous results because of differences in relation to study participants and research methods. Furthermore, we identified some variables with significant effects on the dependent variable through a stepwise regression analysis; these variables had relatively low explanatory power. On the other hand, the strength of this study should be noted. A nationally representative dataset of a large sample size in South Korea was analyzed in this study.

## 5. Conclusions

This study aimed to present foundational data for improving workers’ quality of life and work productivity by exploring factors that affect absenteeism and presenteeism among workers. Health problems due to general fatigue had the greatest impact on presenteeism, followed by the inhalation of smoke, fumes, powder, or dust, exposure to low temperatures indoors/outdoors, health problems due to low back pain, and other health problems. Regarding the factors that affect absenteeism among workers, health problems caused by accidental injuries had the greatest effect, followed by perceived overall health, use of internet and emails for work, working under a strict deadline, inhalation of smoke, fumes, powder, or dust, and finally, presenteeism. Therefore, we confirmed that occupational health and safety are factors that increase workers’ quality of life and productivity and that absenteeism and presenteeism decrease workers’ quality of life.

Thus, it would be important to first improve the factors that were found. Thus, it would be important to first improve the factors that were found to affect absenteeism and presenteeism, such as working conditions and health status. Particularly, a systematic management of presenteeism would be needed to reduce absenteeism. 

## Figures and Tables

**Table 1 ijerph-18-11214-t001:** General Characteristics of the Study Population (*n* = 50,007).

Variable		*n* (%)
Gender	Male	24,943 (49.9)
	Femal	25,064 (50.1%)
Age (years)	15–19	352 (0.7)
	20–29	4523 (9.0)
	30–39	9680 (19.4)
	40–49	13,523 (27.0)
	50–59	12,090 (24.2)
	60≤	9838 (19.7)
Employment status	Self-employed without employees	13,118 (26.2)
	Self-employed with employees	3565 (7.1)
	Employee	31,230 (62.5)
	Unpaid family workers	2062 (4.1)
	Other	30 (0.1)
	DK/no opinion	3 (0.0)
Occupational status	Full-time employee	22,841 (73.1)
	Temporary employee	5817 (18.6)
	Day employee	2356 (7.5)
	DK/no opinion	215 (0.7)
Work	Administrator	416 (0.8)
	Professional	7568 (15.1)
	Office worker	7542 (15.1)
	Service worker	6767 (13.5)
	Sales worker	9206 (18.4)
	Agriculture, forestry and fishery industry skilled worker	4475 (8.9)
	Technical skilled worker and related skilled worker	4015 (8.0)
	Equipment machinery operator and assembly worker	4698 (9.4)
	Simple labor worker	5228 (10.5)
	Soldier	91 (0.2)

**Table 2 ijerph-18-11214-t002:** Variables significantly correlated with absenteeism and presenteeism.

Variable	Q72: Absenteeism	Variable	Q74: Presenteeism
	Correlation Coefficient		Correlation Coefficient
KQ4	0.097 **	KQ4	−0.021 **
Q18	−0.042 **	Q18	−0.117 **
Q23-A	−0.052 **	Q20	−0.108 **
Q23-C	−0.078 **	Q23-A	0.035 **
Q23-D	−0.052 **	Q23-B	0.037 **
Q23-E	−0.081 **	Q23-C	0.077 **
Q23-G	−0.033 *	Q23-D	0.030 **
Q24-A	−0.062 **	Q23-E	0.051 **
Q24-C	−0.050 **	Q23-F	0.011 *
Q24-F	0.074 **	Q23-G	0.033 **
Q24-G	0.052 **	Q23-H	0.050 **
Q24-H	0.114 **	Q24-A	0.154 **
Q24-I	0.109 **	Q24-B	−0.014 **
Q30	0.045 **	Q24-C	0.076 **
Q33	0.058 **	Q24-D	0.088 **
Q36	0.060 **	Q24-E	0.105 **
Q45-A	0.038 *	Q24-F	0.074 **
Q45-B	0.046 **	Q24-G	0.027 **
Q66	−0.035 *	Q24-H	−0.067 **
Q67	−0.035 *	Q24-I	−0.059 **
Q68	0.162 **	Q30	−0.046 **
Q69-A	−0.043 **	Q32	−0.032 *
Q69-C	−0.069 **	Q33	−0.045 **
Q69-D	−0.048 **	Q36	−0.032 **
Q69-E	−0.070 **	Q44-A	0.061 **
Q69-H	−0.035 *	Q44-B	0.068 **
Q69-I	−0.054 **	Q45-A	0.068 **
Q69-J	−0.137 **	Q45-B	0.053 **
Q69-K	−0.036 *	Q66	0.102 **
Q69-M	−0.035 *	Q67	0.080 **
Q69-N	−0.050 **	Q68	−0.183 **
Q74	0.034 *	Q69-A	0.057 **
		Q69-B	0.088 **
		Q69-C	0.201 **
		Q69-D	0.226 **
		Q69-E	0.210 **
		Q69-F	0.166 **
		Q69-G	0.063 **
		Q69-H	0.049 **
		Q69-I	0.048 **
		Q69-J	0.092 **
		Q69-K	0.073 **
		Q69-L	0.190 **
		Q69-M	0.101 **
		Q69-N	0.031 **
		Q72	0.034 *

* *p* < 0.05, ** *p* < 0.01.

**Table 3 ijerph-18-11214-t003:** Factors that affect presenteeism.

Variable	B	Partial R^2^	Model R^2^	F	*p*
Presenteeism (Constant)	2.276				
Q69-L	0.214	0.077	0.077	32.201	<0.001
Q23-E	0.124	0.021	0.098	21.299	<0.001
Q23-D	0.058	0.017	0.116	17.324	<0.001
Q69-C	0.135	0.009	0.125	14.417	<0.001
Q69-N	−0.981	0.009	0.134	12.531	<0.001

**Table 4 ijerph-18-11214-t004:** Factors that affect absenteeism.

Variable	B	Partial R^2^	Model R^2^	F	*p*
Absenteeism (Constant)	14.534				
Q69-J	−7.595	0.031	0.031	53.267	<0.001
Q68	1.115	0.007	0.038	33.345	0.003
Q24-I	0.330	0.003	0.041	24.431	0.035
Q45-B	0.353	0.002	0.043	19.589	0.007
Q23-E	−0.512	0.003	0.046	17.041	0.008
Q74	1.115	0.002	0.048	14.913	0.043

## Data Availability

The Korean version of the Working Conditions Survey (KWCS) is publicly available at http://www.kosha.or.kr/ under permission from Korea Occupational Safety and Health Agency (KOSHA).

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
