# Peer review of "Exploratory Analysis of Related Factors with Absenteeism and Presenteeism on Workers: Using the Fourth Korea Working Condition Survey"

_ijerph, 2021, doi:10.3390/ijerph182111214_

Round 1
Reviewer 1 Report
In first, I would like to thank the authors for the work done and the opportunity to review the data presented by them.
The manuscript entitled “Exploratory Analysis of Related Factors with Absenteeism and Presenteeism on Workers: Using The 4th Korea Working Condition Survey" has great potential, but I recommend that some profound improvements be made.
- Introduction: in this section the authors only refer to the negative consequences of absenteeism and presentism for organizations, there is no reflection or theorization about the antecedents. But antecedent is the data analysis focus.
- Method: It is not clear what the theoretical dimensions of the questionnaire are. In this section I congratulate the authors on the size and representativeness of the sample.
- Results and discussion: No data were presented on the reliability of the scales used. There is no reflection or theorization about the same item or factor has a positive relationship with absenteeism and negative with presentism and vice versa. The discussion section begins with the presentation of results from previous studies when it should be reflecting the data themselves. Additionally, as an exploratory study, it should reflect on implications for future studies. To affirm that the conditions that precede presentism and absenteeism are necessary better seems to me insufficient for the practical implications of the studies.
Author Response
- Introduction: in this section the authors only refer to the negative consequences of absenteeism and presentism for organizations, there is no reflection or theorization about the antecedents. But antecedent is the data analysis focus.
Insert in introduction according to review opinion below:
Previous studies have indicated several determinants of presenteeism including organizational factors as understaffing, low organizational support, and strict absence policy and work‐related factors as workload and time pressure, as well as individual factors as affective commitment and financial difficulties [6]. High prevalence of presenteeism is concerning, considering that it could lead to health problems and absenteeism in later period [7]. Previous studies reported that the experience of presenteeism was related to poor self‐rated health, depression, absenteeism, and lower work performance [8-11].
- Method: It is not clear what the theoretical dimensions of the questionnaire are. In this section I congratulate the authors on the size and representativeness of the sample.
Insert or changed in method part according to review opinion below:
- The survey population was a nationally representative sample of an economically active population (≥15 years old) including waged workers, self‐employed/employer, unpaid family workers, and others.
- The KWCS is publicly available at http://www.kosha.or.kr/ under permission from Korea Occupational Safety & Health Agency.
- The size of this study population was 50,007 who workers participated in the on-site survey about working and employment conditions.
- Results and discussion: No data were presented on the reliability of the scales used. There is no reflection or theorization about the same item or factor has a positive relationship with absenteeism and negative with presentism and vice versa. The discussion section begins with the presentation of results from previous studies when it should be reflecting the data themselves. Additionally, as an exploratory study, it should reflect on implications for future studies. To affirm that the conditions that precede presentism and absenteeism are necessary better seems to me insufficient for the practical implications of the studies.
Insert or changed in method part according to review opinion below:
1. In the 2014 KWCS, medical absenteeism was 8.8%, while presenteeism was 25.8% [5].
2. On the other hand, the strength of this study should be noted. A nationally representative dataset of a large sample size in South Korea was analyzed in this study.
3. Therefore, we confirmed the factors of occupational health and safety for increasing of workers’ quality of life and productivity and decreasing of absenteeism and presenteeism.

Reviewer 2 Report
The paper is interesting and well structured. However, before publication, a series of improvements should be done.
- More details about the stepwise regression analysis should be provided: is it a forward stepwise procedure or a backward procedure?
- Which of several criteria at the disposal of a researcher was used to select between different models?
- Tables 2 and 3 should be redesigned because they are difficult to understand. What is model R2? The R2 when all variables are included?
- I recommend to present a simple equation of the best model selected by your procedure and show that it passes all the tests required for a linear regression model.
- The data used in this study should be presented with more details, for example some descriptive statistics etc.
- Emphasize what is the new knowledge this paper brings.
- Present a comparison with other similar studies and show the strengths of this study.
Author Response
- More details about the stepwise regression analysis should be provided: is it a forward stepwise procedure or a backward procedure?
=> From to https://en.wikipedia.org/wiki/Stepwise_regression
Stepwise regression is the step-by-step iterative construction of a regression model that involves the selection of independent variables to be used in a final model. It involves adding or removing potential explanatory variables in succession and testing for statistical significance after each iteration.
So I did selected the stepwise regression as bidirectional elimination than forward selection(regression) and backward elimination(regression).
Which of several criteria at the disposal of a researcher was used to select between different models?
Response to review below:
Because, stepwise regression is a method of fitting regression models in which the choice of predictive variables is carried out by an automatic procedure. In each step, a variable is considered for addition to or subtraction from the set of explanatory variables based on some prespecified criterion.
Tables 2 and 3 should be redesigned because they are difficult to understand. What is model R2? The R2 when all variables are included?
Response to review below:
In statistics, the coefficient of determination, also spelt coëfficient, denoted R2 or r2 and pronounced "R squared", is the proportion of the variation in the dependent variable that is predictable from the independent variable(s).
From to https://en.wikipedia.org/wiki/Coefficient_of_determination
I recommend to present a simple equation of the best model selected by your procedure and show that it passes all the tests required for a linear regression model.
Response to review below:
In standard multiple regression all predictor variables are entered into the regression equation at once. But in a stepwise regression, predictor variables are entered into the regression equation one at a time based upon statistical criteria.
The data used in this study should be presented with more details, for example some descriptive statistics etc.
Insert the Table 1 in paper.
Emphasize what is the new knowledge this paper brings.
Insert in paper below:
Therefore, we confirmed the factors of occupational health and safety for increasing of workers’ quality of life and productivity and decreasing of absenteeism and presenteeism.
Present a comparison with other similar studies and show the strengths of this study.
Insert in paper below:
On the other hand, the strength of this study should be noted. A nationally representative dataset of a large sample size in South Korea was analyzed in this study.

Round 2
Reviewer 2 Report
The author addressed the issues raised in the previous round of review providing enough information.